

**Technical note.**
**Harmonization of the multi-scale multi-model activities HTAP, AQMEII and**
**MICS-Asia: simulations, emission inventories, boundary conditions and**
**output formats**
Stefano Galmarini[1], Brigitte Koffi[1], Efisio Solazzo[1], Terry Keating[2], Christian Hogrefe[3],
Michael Schulz[4], Anna Benedictow[4], Jan Jurgen Griesfeller[4], Greet Janssens-Maenhout[1],
Greg Carmichael[5], Joshua Fu[6], Frank Dentener[1]
1. European Commission, Joint Research Centre, Ispra, Italy
2. Environmental Protection Agency, Applied Science and Education Division, National Center for
Environmental Research,  Office of Research and Development, Headquarters, Federal Triangle, DC 20460 DC,
USA
3. Environmental Protection Agency, Computational Exposure Division, National Exposure Research
Laboratory, Office of Research and Development, Research Triangle Park, NC 27711, USA,
4. Norwegian Meteorological Institute, Oslo, Norway
5. Center for Global and Regional Environmental Research, University of Iowa, Iowa City, IA 52242, USA
6. Department of Civil & Environmental Engineering, The University of Tennessee, Knoxville, TN 37996, USA
**Abstract**
We present an overview of the coordinated global numerical modelling experiments
performed during 2012-2016 by the Task Force on Hemispheric Transport of Air Pollution
(TF HTAP), the regional experiments by the Air Quality Model Evaluation International
Initiative (AQMEII) over Europe and North America, and the Modelling Intercomparison
Study- Asia (MICS-Asia). To improve model estimates of the impacts of intercontinental
transport of air pollution on climate, ecosystems and human health and to answer a set of
policy relevant questions, these three initiatives performed emission perturbation modelling
experiments consistent across the global, hemispheric and continental/regional scales. In all
three initiatives, model results are extensively compared against monitoring data for a
range of variables (meteorological, trace gas concentrations, and aerosol mass and
composition) from different measurement platforms (ground measurements, vertical
profiles, airborne measurements) collected from a number of sources. Approximately 10 to
20 modelling groups have contributed to each initiative, and model results have been
managed centrally through three data hubs maintained by each initiative. Given the
organizational complexity of bringing together these three initiatives to address a common
set of policy relevant questions, this publication provides the motivation for the modelling
activity, the rationale for specific choices made in the model experiments, and an overview
of the organizational structures for both the modelling and the measurements used and
analysed in a number of modelling studies in this special issue.

**1. Introduction**



The Task Force on Hemispheric Transport of Air Pollution (TF HTAP) was organized in 2005
under the *UNECE Convention on Long-range Transboundary Air Pollution* (CLRTAP) (see
http://www.unece.org/env/lrtap/welcome.html). Recognizing the increasing importance of
hemispheric transport of air pollution, CLRTAP mandated the TF HTAP to work in
partnership with scientists across the world to improve knowledge on the intercontinental
or hemispheric transport and formation of air pollution; its impacts on climate, ecosystems,
and human health; and the potential mitigation opportunities.
In 2010, TF HTAP produced the first comprehensive assessment of the intercontinental
transport of air pollution in the Northern Hemisphere (TF HTAP, 2010a,b). A series of four
reports addressed issues around emissions, transport, and impacts of particulate matter and
ozone, mercury, POPs, and their relevance for policy. The HTAP Phase 1 (HTAP1) joint
modelling experiments, in which more than 20 global models participated, focussed on the
meteorological year 2001. In 2012, the TF HTAP launched a new phase of cooperative multi-
model experiments and analyses to provide up-to-date information to CLRTAP (e.g. Maas
and Grenfellt, 2016) and other multi-lateral cooperative efforts, as well as national actions
to decrease air pollution and its impacts.
The objectives of the HTAP Phase 2 (HTAP2) activity are summarized as follows:
• To estimate relative contributions of regional and extra-regional sources of air
pollution in different regions of the world, by refining the source/receptor
relationships derived from the HTAP Phase 1 simulations.
• To provide a basis for model evaluation and process studies to characterize the
uncertainty in the estimates of regional and extra-regional contributions and
understand the differences between models.
• To give input to assessments of the impacts of control strategies on the contribution
of regional and extra-regional emissions sources to the exceedance of air quality
standards and to impacts on human health, ecosystems, and climate.
The major advances of HTAP2 over the earlier HTAP1 experiments were:
• a focus on more recent years as a basis for extrapolation (2008-2010), including an
updated collection of emission inventories for 2008 and 2010 (Janssens-Maenhout et
al., 2015) that is utilised across all model experiments. In HTAP1 the year of interest
was 2001, and in contrast to HTAP2, the anthropogenic emissions used by the
different modelling groups were expected to be loosely representative for the
beginning of the 2000s, but were not prescribed, resulting in a large diversity of
base-line emissions.
• an expanded number of more refined source/receptor regions: the original set of 4
rectangular source regions (North America, Europe, South Asia, and East Asia)
identified in HTAP1 have been refined to align with geo-political borders and
additional regions have been added, dividing the world into 16 potential source
regions and 60 receptor regions.
• the use of regional models and consistent boundary conditions from selected global
models for Europe, North America, and Asia to provide high resolution estimates of
the impacts on health, vegetation, and climate, in addition to the global models'
world-wide coverage.



The most innovative aspect of the modelling work, performed in 2013-2016, is the
consistent coupling of global and regional model experiments using existing modelling
frameworks. The regional counterparts of the TF HTAP are the AQMEII (Air Quality Model
Evaluation International Initiative) and MICS-Asia (Model Intercomparison Study for Asia)
activities.
The AQMEII project was launched in 2008 in an attempt to bring together modelers from
both sides of the Atlantic Ocean to perform joint regional model experiments using common
boundary conditions, emissions, and model evaluation frameworks with a specific focus on
regional modeling domains over Europe and North America (Rao et al., 2012). The first two
AQMEII activities focused on the development of general model-to-model and model-to-
observation evaluation methodologies (phase 1, Galmarini et al. 2012a) and the simulation
of aerosol/climate feedbacks with on-line coupled modeling systems (phase 2, Galmarini et
al. 2015). AQMEII Phase 3 (AQMEII3) is devoted to performing joint modeling experiments
with HTAP2. The AQMEII modeling community includes almost all of the major existing
modeling systems for regional scale chemical transport simulation in research and
regulatory applications in both continents. Most of the groups participating are part of
modeling initiatives in the individual European member states and some of these groups
utilize models developed in North America, thus providing the opportunity of assessing the
impact of users outside of the conventional modeling context.
The MICS-Asia Phase III (MICS3) project is an activity building on work performed in Phase I
(1998-2000; sulphur transport and deposition) and Phase II (2004-2009; sulphur, nitrogen,
ozone and aerosols, see Fu et al., 2008). MICS3 is organized as a multi-national consortium
of institutions and brings together modellers from China, Japan, Korea, Southeast Asia and
the United States. The overall scope of MICS3 includes evaluation of the ability of models to
reproduce pollutant concentrations under highly polluted conditions, dry and wet
deposition fluxes, and the quantification of the effects of uncertainties due to process
representation (emissions, chemical mechanisms, transport and deposition) and model
resolution on simulated air quality. The joint evaluation with HTAP2 focuses on the
evaluation of the role of long-range transport of air pollution in East Asia on air quality and
impacts on climate, ecosystems and human health.
The involved framework for global aerosol modelling is the AeroCom initiative (Aerosol
Comparison between observations and models, Schulz et al. 2009, Myhre et al. 2013), and
dedicated experiments on long-range transport were designed and performed in
collaboration with HTAP as part of AEROCOM phase 3 (see
https://wiki.met.no/aerocom/phase3-experiments), with an additional focus on long-range
transport of dust and fire derived aerosol. The data storage and evaluation platform for
global models was shared between AeroCom and HTAP2 (see section 2.5).
Presently these three activities involve ca. 10 global scale models, and approximately thirty
regional scale modelling groups performing model simulations on the North American,
European and East Asian domains, probably making HTAP2/AQMEII3/MICS3 exercise the
largest, multi-scale/multi-model activity ever performed in atmospheric chemical modelling.
The multi-scale and multi-regional modelling exercise required three independent
organizations to manage and engage their respective communities and an overarching
coordination effort as well as a high level of harmonization of the model simulations aiming





at comparability, usability and interoperability of the model results at the various scales.
Specific decisions were made regarding the simulation period, lower air boundary
conditions (emission inventory), volatile organic carbon (VOC) speciation, methane
concentrations, emission perturbation runs, source region perturbations, lateral and upper
air boundary conditions for regional simulations, variables expected for the analysis, file
naming conventions, type and location of monitoring sites where model results were
expected, data submission procedures, and the development and use of interoperable data
archiving and visualisation servers.
The scope of this note is to provide information on the modelling activity harmonization and
coordination adopted to guarantee the maximum level of coherence between the global
and regional simulations. It will provide specific details on the organization of the global
HTAP2 and the regional AQMEII3 activities, while only general information on the MICS3
experiments is provided. Additional details regarding HTAP2 are summarised at
http://iek8wikis.iek.fz-juelich.de/HTAPWiki/ and are available in the report by Koffi et al.
(2016) and for AQMEII3 at http://ensemble2.jrc.ec.europa.eu/aqmeii/.
This note should serve to provide coherent information on the simulations performed and
their characteristics to the analysis articles presented in this special issue.
**2. The HTAP2, AQMEII3, and MICS3 modelling exercises set up**
The following aspects have to be harmonized in the organization of a multi scale multi
chemical transport model activity:

-  Simulation periods and meteorology to be used
-  Emission inventories for global and regional models
-  Boundary conditions for regional scale air quality models
-  Harmonisation and interoperability of global and regional model output
-  Monitoring data locations and methods for comparing models with observations
-  Documentation of individual model set-up and construction of ensemble averages.
**2.1 Simulation period and meteorology used**
The simulation period of interest 2008-2010 was chosen on the basis of the availability of
emissions data and intensive observations. The models were requested to run the three-
year period with a priority given to the year 2010, followed by 2008, and then 2009. Global
models can use meteorological data representative of the respective year, e.g. driven or
constrained by one of the global analysis products that were most convenient to the
modelling group.  Regional scale modellers also were free to use the meteorological model
of their choice based on compatibility with their chemical transport model. Sets of chemical
boundary conditions for the regional models were provided by a limited set of global
models participating in the global modelling experiments (see section 2.4)





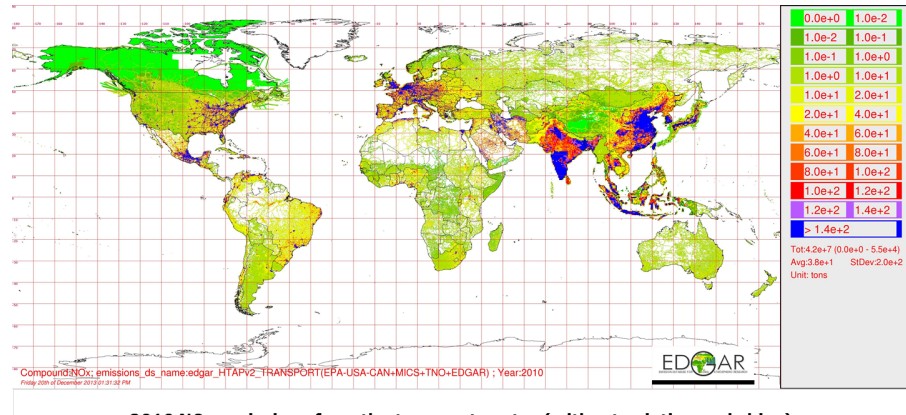

**2010 NOx emissions from the transport sector (without aviation and ships)**

**Figure 1**. Example of HTAP_v2.2 emission mosaics for NO$_x$ in the transport sector.

## 2.2 Emission data

The anthropogenic emission data were harmonized across the regional and global modelling
experiments. The Joint Research Centre's (JRC) EDGAR (Emission Data Base for Global
Researc) team, in collaboration with regional emission experts from the U.S. Environmental
Protection Agency (EPA), EMEP (European Monitoring and Evaluation Programme, CEIP
(Centre on Emission Inventories and Projections), TNO (Netherlands Organisation for
Applied Research), the MICS-Asia Scientific Community and REAS (Regional Emission Activity
Asia), has compiled a composite of regional emission inventories with monthly gridmaps
that include EDGARv4.3 gap filling for regions and/or sectors that were not provided by the
regional inventories.
The so-called HTAP_v2.2 database (Janssens-Maenhout et al., 2015), used in the global
modelling experiments, has the following characteristics:
• Years 2008 and 2010, yearly and monthly time resolutions
• Components: SO$_2$, NO$_x$, NMVOC, CH$_4$, CO, NH$_3$, PM$_{10}$, PM$_{2.5}$, BC, and OC at sector-
specific level.
• 7 emission sectors (Janssens-Maenhout et al., 2015), see Table 1.
• Global geo-coverage with spatial resolution of 0.1° x 0.1° longitude, and latitude, to
serve the needs of both global and regional model activities.

Annual gridded emission data (http://edgar.jrc.ec.europa.eu/htap_v2, latest access July,
2016) are delivered for each pollutant and emission sector.  Monthly gridded values are
provided for some sectors (energy, industry, transport and residential), where information
was available to disaggregate annual emissions.
The regional emissions for the North American and European regional scale simulations of
AQMEII3 are described in Pouliot et al. (2015), and were used earlier for AQMEII2 (Galmarini





et al., 2015) and embedded into the HTAP_v2.2 inventory. The Asian inventory MIX (Li et
al., 2015) was developed for MICS3 and HTAP2 simulations on a 0.25°x0.25° resolution, and
converted by raster resampling to 0.1°x0.1° resolution for use in HTAP2 . These regional
inventories have been combined to form a global mosaic (**Figure 1**) that is consistent with
inventories used at the regional scale in Europe, North America and Asia. However, we note
that these emission estimates stemming from different data sources for different regions of
the world, are not necessarily consistent, for example different fuel statistics or emission
factors may have been used for different regions. Details on the recommended VOC
speciation and other specific emission information can be found in Koffi et al. (2016),
Janssens Maenhout (2015), Li et al. (2015) and Pouliot et al. (2015).
**Table 1**: Emission sectors in HTAP_v2.2 database

| Sector | Description |
| --- | --- |
| AIR | International and domestic aviation |
| SHIPS | International shipping |
| ENERGY | Power generation |
| INDUSTRY | Industrial non-power large-scale combustion emissions and emissions of industrial processes and product use including solvents |
| TRANSPORT | Ground transport by road, railway, inland waterways, pipeline and other ground transport of mobile machinery. Does not include re-suspension of dust from pavements or tire and brake wear |
| RESIDENTIAL | Small-scale combustion, including heating, cooling, lighting, cooking and auxiliary engines to equip residential and commercial buildings, service institutes, and agricultural facilities and fisheries; solid waste (landfills/ incineration) and wastewater treatment |
| AGRICULTURE | Agricultural emissions from livestock, crop cultivation but not from agricultural waste burning and not including savannah burning |

Biomass burning emissions have not been prescribed for the global modelling groups, but it
is recommended that groups use GFED3 data, which are available at daily and 3-hour
intervals (see http://globalfiredata.org/). For the regional modelling groups participating in
AQMEII3, fire emissions were included in the inventories distributed to the participants
(Pouliot et al., 2015; Soares et al., 2015). Biogenic NMVOCs, soil and lightning $NO_x$, dust, and
sea salt emissions have not been prescribed for either the global or regional modelling





groups; modelling groups are encouraged to use the best information that they have available except that the AQMEII3 regional modelling groups were advised not to include lightning $NO_x$ in their simulations since not all modelling groups had a mechanism for including them. For wind-driven DMS (dimethyl sulphide) emissions from oceans, the climatology of ocean surface concentrations described in Lana et al. (2011) was recommended in conjunction with the model's meteorology and emission parameterisation for the global models. The regional models participating in AQMEII3 did not consider DMS emissions. For volcanic emissions, it was recommended that global groups use the estimates developed for 2008-2010 for AeroCom as an update of the volcanic $SO_2$ inventory of Diehl et al. (2012) and accessible at http://aerocom.met.no/download/emissions/HTAP/ (latest access July 2016). As in the case of lightning $NO_x$ emissions, the AQMEII3 regional modelling groups were advised not to include volcanic emissions in their simulations since not all modelling groups had a mechanism for including them. Modeling groups were asked to document the source of all of their emissions data and assumptions, especially if it deviated from the recommended parameterisations. For mercury, the AMAP/UNEP global emissions inventory for 2010 was recommended (http://www.amap.no/mercury-emissions). None of the regional models participating in AQMEII3 considered mercury in their simulations.

**2.3 Emission perturbation**

In addition to the base 2008-2010 simulations, modelling groups were requested to perform emission perturbation experiments to help estimate source/receptor relationships; to attribute estimated concentrations, depositions, and derived impacts to regional and extra-regional sources; and to be used for scenario evaluations including uncertainties. **Figure 2** lists a large number of possible perturbation experiments; all except the methane perturbation experiments involve a 20% decrease in anthropogenic emissions similar to HTAP1. The choice of 20% was motivated by the consideration that the perturbation would be large enough to produce a sizeable impact (i.e. more than numerical noise) even at long-distances, while small enough to be in the near-linear atmospheric chemistry regime, assumptions which are subject to further analysis. The emission decreases are specified for combinations of pollutants, regions, and sectors.



PM = Other Particulate Matter (BC, OC, PM10, PM2.5)
TRN = Ground Transport Sector; PIN = Power and Industry Sectors; RES = Residential Sector; OTH = Other Sectors (Ships, Aviation, Agriculture); FIR = Fire
DST = Dust * For dust, some models should divide the NAF source into separate source regions for the Sahara (091+092, in the Tier2 regions) and Sahel (093).

**Figure 2**. HTAP2 emission perturbation experiments, dark green color are highest priority experiments, light green next priority, and white colors lower priority. ALL refers to perturbation of all anthropogenic components and sectors, sectors are TRN (Transportation), PIN (Power+industry), RES (Residential), OTH (Other), FIR (Fire), DST (Mineral dust).

To capture the impact of changing methane emissions in a single year simulation, it is necessary to perturb the methane concentration instead of the emissions. The recommended perturbations (Table 2) are intended to cover the range of $CH_4$ concentration changes associated with the Representative Concentration Pathway (RCP) scenarios used for the Intergovernmental Panel on Climate Change (IPCC) fifth assessment report (AR5) (IPCC, 2013) for 2030. The highest priority was assigned to an increase of global $CH_4$ concentrations to 2121 $ppb_v$ (representative of RCP8.5). The next priority is assigned to a decrease of global $CH_4$ concentrations to 1562 $ppb_v$ (representative of RCP2.6).

Table 2: BASE and Methane Perturbation runs

| Simulation | Global $CH_4$ Concentration (ppbv) | Representative of |
|---|---|---|
| BASE | 1798 | 2010 based on IPCC (2013) |
| CH4INC | 2121 | 2030 under RCP 8.5 |
| CH4DEC | 1562 | 2030 under RCP2.6 |

The combination of global (all regions and sources) and regional perturbation experiments provides the necessary information to calculate the so-called RERER (Response to Extra-Regional Emission Reductions) metric, using the information on the contribution of foreign





emission perturbations relative to all worldwide emission perturbation to a change in region
i.

$$RERER_i = \frac{\Sigma R_{foreign}}{\Sigma R_{all}} = \frac{R_{global} - R_{region,i}}{R_{global}}$$
(eq 1)

where $R_{global}$ is calculated using the global (all regions and sources) 20% perturbation
simulation (GLO) minus the unperturbed simulation (BASE) and $R_{region}$ is the corresponding
difference of the regional 20% emission perturbation simulation and the base simulation.
The metric can be applied to a range of quantities, including surface concentrations, column
amounts, and derived parameters.
A low (i.e. near 0) RERER value means that the signal within a region is not very sensitive to
extra-regional emission reductions, and that local concentrations (or column amounts, etc.)
depend more on local emission reductions given the current distribution of anthropogenic
and biogenic emissions.  A high RERER value (i.e. near 1) suggests that local conditions are
strongly influenced by emissions changes outside the region. In some circumstances, when
emission reductions correspond to increasing concentrations (e.g. ozone titration by NO
emissions), RERER can become larger than 1.





### 2.4 Boundary Conditions for Regional Simulations

One of the new aspects of HTAP2 experiments is the coupling of global and regional model simulations, including coupled emission perturbation studies. These common experiments are intended to enable the examination of the effects of a) the finer spatial and temporal resolution of regional models and b) the different processes represented in global and regional models.

In order to "nest" the regional within the global simulations, computational results from one or more global models are needed as boundary conditions for the regional models' domains (**Figure 3**), typically provided as a set of time-varying concentrations of medium-to-long-lived components in a 3D box over the respective regional model domains at typical time resolutions of 3 to 6 hours.

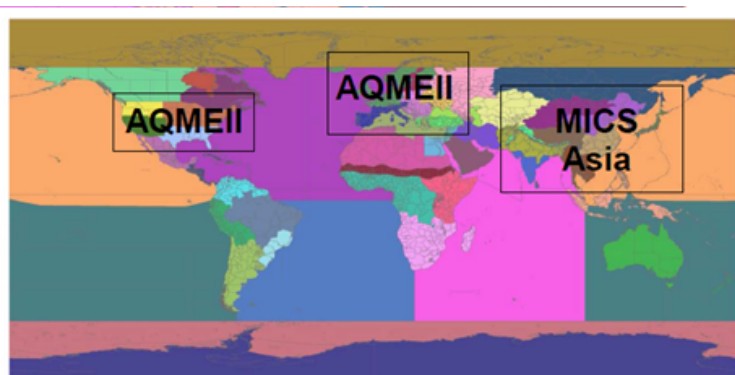

**Figure 3**: Domains of the regional model simulations and source receptor areas

A small number of the global models participating in HTAP2 provided boundary conditions for regional simulations, the choice depending mostly on existing experiences of regional communities with these particular global models. The global scale simulations that were made available to the regional scale modelers for defining boundary conditions are presented in Table 3. Boundary conditions were provided for both the base case and also for a number of emission perturbation runs. Each of the emissions perturbation experiments with the global models created a new set of boundary conditions that can be used at the regional scale. This nesting is depicted graphically in Figure 4. It shows an example where the HTAP2 source region (in this case, East Asia) is wholly within the regional model domain. The inclusion of the global perturbation simulation (GLOBALL) allows consistent evaluation of the RERER metric (see section 2.3).





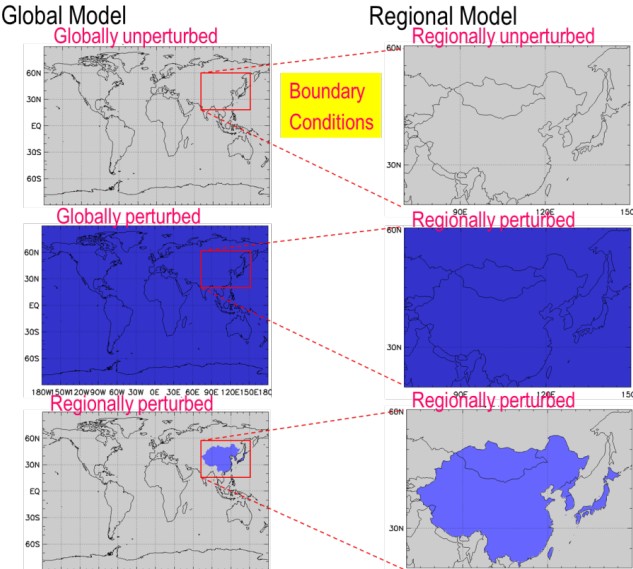

**Figure 4**: Example set of experiments, with both global and regional model (in this case a
regional model over East Asia, red box), where the regional source perturbation is East Asia
(blue shading), and is wholly within the regional model domain. Note that the magnitude of
the emission perturbation in the region of consideration is identical between the global and
regional model.
Regional models where free to use as boundary conditions one or more models as
long as they were selected from the set of global models participating in HTAP2
(Table 3), but in practice the AQMEII3 community focused its effort on C-IFS(CB05)
(Flemming et al.,2015) calculations.  GFDL/AM3 (Lin et al, 2012a,b) and GEOS-Chem
(Park et al., 2004, Bey et al., 2001) were additionally used in some North American
simulations. GEOS-Chem and CHASER (Sudo et al., 2002; 2007, Watanabe et al.,
2011, Sekiya and Sudo, 2014) were the preferred models for the MICS3 consortium.



**Table 3**: 2008, 2009 and 2010 HTAP2 Global Runs for Regional Boundary Conditions

| Model | Spatial Resolution | Temporal Resolution | Chemistry | Simulations |
|---|---|---|---|---|
| **C-IFS(CB05) (ECMWF)** | 1.125°x1.125° (T159) 54 levels | 3 hourly | CB05 | **BASE GLOALL CH4INC NAMALL EURALL EASALL SASALL** |
| **GFDL/AM3** | ~1°x1° 48 levels | 3 hourly | | **BASE GLOALL CH4INC NAMALL EURALL EASALL** |
| **GEOS-Chem** | 2.5°x2° 47 levels | 3 hourly | | **BASE GLOALL CH4INC NAMALL EURALL EASALL** |
| **CHASER** | 2.8°x2.8° | 3 hourly + daily mean | | **BASE** |

**2.5 Specification of the global and regional scale model outputs**
Careful consideration was given to the organization of the model output, given the
large number of models, variables requested, and case studies. This required
specifications of data formats, variable and file naming conventions, data
organization at identified collection points, and the definition of agreed locations
where measurements would be available and model data had to be produced for
both regional and global models. Further details can be found at
http://iek8wikis.iek.fz-juelich.de/HTAPWiki/HTAP-2-data-submission and in Koffi et
al. (2016). For HTAP2 and AQMEII3, the experience acquired over the past
experiments allowed this massive data handling task to be carried out in an efficient
way because data formats, naming conventions and collections points were already
well established for these two activities and respective communities of models. For
HTAP2 the netCDF (http://www.unidata.ucar.edu/software/netcdf/) with Climate





and Forecast (CF) (http://cfconventions.org/) meta data format was adopted. For
AQMEII3 the ENSEMBLE data format was used (Galmarini et al. 2012b), allowing easy
participation for regional modellers already participating in AQMEII2. Two data
repositories were available for the two communities: the AeroCom repository at the
Norwegian meteorological institute (MetNo) (aerocom.met.no; Schulz et al., 2009)
and the JRC ENSEMBLE (Galmarini et al., 2014) platforms, respectively.   Data for
MICS3 were handled and analyzed at the Joint International Center on Air Quality
Modeling Studies (JICAM) in Beijing, China, a joint cooperation between the Institute
of Atmospheric Physics (IAP) of Chinese Academy of Sciences and the Asia Center for
Air Pollution Research (ACAP) in Niigata, Japan. These facilities not only allow the
organization of the data produced by various sources around the world but also their
consultation through web interfaces and the matching of the model results with the
available measured data and the statistical comparison of these two pieces of
information. A connection and automatic data conversion protocol between the
ENSEMBLE and AeroCom platforms was also pioneered to allow the bi-directional
transfer of model data and a consistent comparison of global and regional model
results with a common set of observations.
Global model data from this study can be accessed via the AeroCom data server at
MetNo. Data are organised such that the HTAP2 model version, experiment, period,
and variable name can be identified readily from directory and file names. Model
output providers have to register at the database provider MetNo and are provided
with    access    to    a    linux    server    via    ssh    (see    further    details    at
https://wiki.met.no/aerocom/user-server). This server also provides essential and
standard data inspection, analysis and extraction tools for netCDF files (ncdump,
ncview, python, nco, cdo, etc.). Users may utilize these tools to retrieve  files, or
subsets of them for further analysis. All incoming files are processed with the
AeroCom visualization tools to generate "quick look" images for initial inspection. All
variables are plotted as fields for major regions, each month and season. Where
available, comparisons are made to surface observations, mainly those from the
EBAS    database    maintained    by    NILU    (ebas.nilu.no)    and    from    Aeronet
(http://aeronet.gsfc.nasa.gov). The quick look images are publicly available via the
web                interface                at                http://aerocom.met.no/cgi-
bin/aerocom/surfobs_annualrs.pl?PROJECT=HTAP&MODELLIST=HTAP-phaseII-ALL.
To facilitate the comparability of model results with measured data, the former were
requested as time series at surface locations, or vertical profiles, mostly located in
Europe and North America, enabling the comparison of the AQMEII3 and HTAP2
experiments.   Model results were requested in various forms. Specifically, 4128
surface stations were identified for the comparison of gas phase species, 2068
surface stations were identified for the comparison of aerosol species, and 240
stations were identified for the evaluation of vertical profiles. These locations are a



mixture of stations of global and regional significance and spatial representativeness
(Figure 5). Details of the data requests for HTAP2 can be found in Koffi et al. (2016).
For AQMEII3, the specifications of requested model variables are contained in the so
called          AQMEII          overarching          document
(http://ensemble2.jrc.ec.europa.eu/aqmeii/?page_id=527). Model results are also
available to participating modelling groups and the wider scientific community
through the ENSEMBLE web based platform following the protocol established for
phase 1 and 2 of AQMEII (Galmarini and Rao, 2011)
MICS3   output   includes   monthly   averaged   hourly   surface   data   for $O_3$,
NO, $NO_2$, $HNO_3$ and HONO; surface VOC species consistent with the CB05, CBMZ,
RADM2 and SAPRC99 mechanisms and Wet/Dry depositions of sulfur and nitrogen
components.
To help diagnose the differences between models and isolate different transport
processes, we requested that HTAP2 global models also include two passive tracers.
These tracers should be emitted in the same quantity as total anthropogenic CO
emissions (not including fires) and decay exponentially with uniform fixed mean
lifetimes (or e-folding times) of 25 and 50 days, respectively, as in the Chemistry-
Climate Modelling Initiative (CCMI).



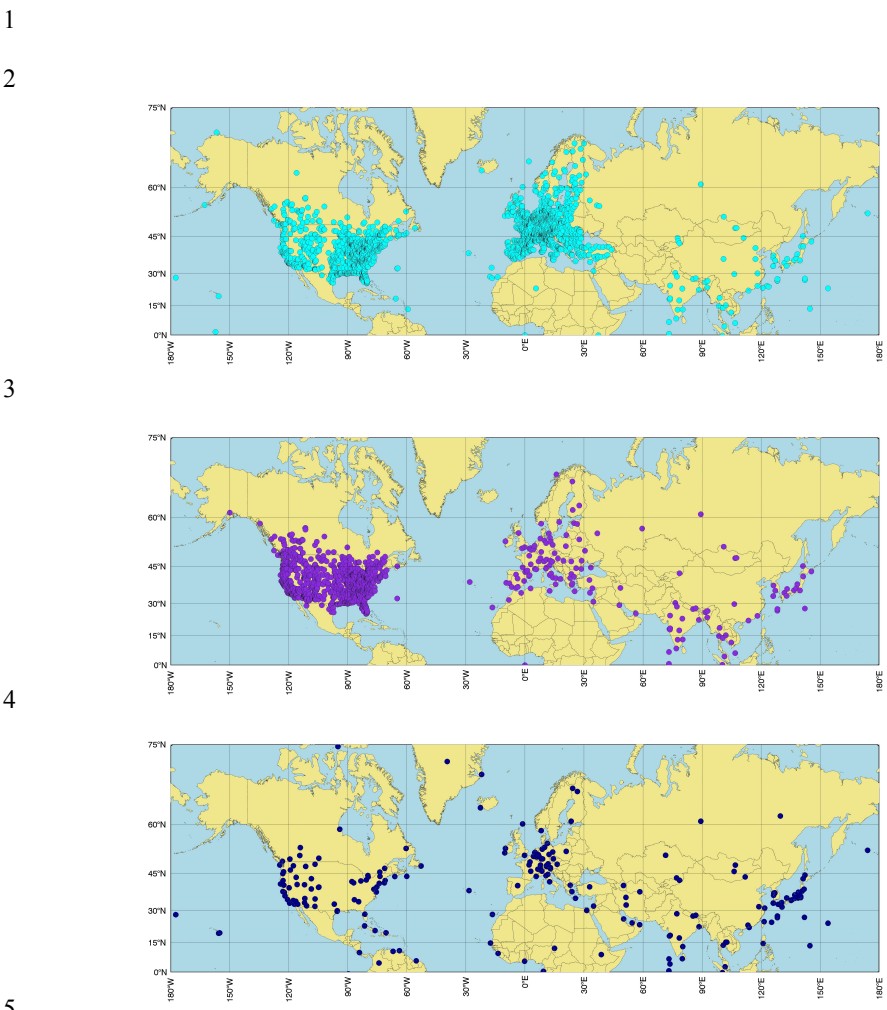

6       **Figure 5**: Location of the stations where surface gas (top), surface aerosol (middle) and
7                              vertical profile (bottom) model outputs are requested.

**3. Conclusions**
This technical note provides details about the set up of the joint regional-global
chemistry-transport emission perturbation experiments, planned and executed
within the HTAP2 model exercise. The Task Force Hemispheric Transport Air
Pollution falls under the *UNECE Convention on Long-range Transboundary Air*
*Pollution* and deals with the *increasingly* important issue of hemispheric transport of
air pollution. TF HTAP works in partnership with scientists across the world to
improve our understanding of the intercontinental or hemispheric transport and





formation of air pollution; its impacts on climate, ecosystems, and human health;,
and the potential mitigation opportunities.
The major advances of HTAP2 with respect to previous HTAP1 activity are:
• a focus on more recent years as a basis for extrapolation (2008-2010),
• a larger number of source/receptor regions
• In collaboration with the existing regional scale modelling initiatives AQMEII
and MICs-ASIA: the use of regional models and consistent boundary
conditions from selected global models for Europe, North America, and Asia
to provide higher resolution estimates of the impacts of hemispheric
transport of air pollution on health, ecosystems and climate.
The multi-model, multi-scale, and multi-pollutant character of the activities
performed in HTAP2 required a considerable level of harmonization of the
information used to run the model at different scales and of the results produced.
Such harmonization considerably facilitates the interpretation of model results and
inter-model differences. Particular attention was given to providing coherent
emissions and boundary conditions to the global and regional scale models, and
harmonising dataset of monitoring data collected to evaluate the model results. To
our knowledge such an attempt is unprecedented in the field and constitutes an
important starting point for future multiple scale modelling activities. A considerable
effort has been made for the harmonization of data formats, and web based data
hubs, allowing consultation of model and measurement data by the participants as
well as possible external data users with simplicity and having a few "one-stop
shops," where all information is collected geo-referenced and ready to be used. As
independently demonstrated in the past, by the ENSEMBLE and AeroCom
experiences, such an approach effectively takes away the burden on individual
modelling groups of collecting scattered measurement data, and organizing these
data sets for comparison with models. Moreover, this approach effectively provides
benchmark datasets for objective comparisons across models.
While first steps towards fuller integration of protocols, requested outputs, and
analysis methods were shared across the three communities, a fully interoperable
and harmonised set of global and regional outputs was not yet obtained due to
different requirements of the communities. At this stage, the availability of global
and regional model outputs and observations at a common set of monitors permits a
first analysis of global/regional model performance in the North American and
European domains and represents a significant step forward for both communities.
Many of the analyses presented in this special issue draw upon this unique collection
of data and tools which is open and available for further analysis.  We encourage the
scientific community to continue to explore this data to generate scientific and



policy-relevant insights and to engage in the future development of the TF HTAP,
AQMEII, and MICS-Asia activities.
**Acknowledgements**
The AeroCom database at MetNo received support from the LRTAP convention
under the EMEP programme, through the service contract to the European
commission no. 07.0307/2011/605671/SER/C3, and benefitted from the Norwegian
research council project #229796 (AeroCom-P3). JRC received support for this work
via Administrative Arrangement AMITO and AMITO2 from the European Commission
DG Environment. TF HTAP, AeroCom, AQMEII, and MICS-Asia exist due to the
relentless contributions of numerous excellent scientists. Although this work has
been reviewed and approved for publication by the U.S. Environmental Protection
Agency, it does not necessarily reflect the views and policies of the agency. We thank
Dr Mian Chin for her support in designing and promoting the HTAP-AEROCOM
experiments.





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
