# Peer review of "Technical note."

_Atmospheric Chemistry and Physics, 2016_

## Referee Comment (RC1) · D.S. Stevenson (Referee) · 11 Nov 2016

**Review of Technical Note: 'Harmonization of the multi-scale multi-model activities of HTAP, AQMEII and MICS-Asia: simulations, emission inventories, boundary conditions and output formats', by Galmarini et al.**

**General Comments**

This overview of the HTAP2, AQMEII and MICS-Asia modelling is a very useful and necessary technical description to accompany/precede the Special Issue papers. My specific comments below are mainly minor suggestions to clarify the text, with a few requests for additional material. The slightly more substantive comments concern the definition of the RERER metric and some of the descriptions of how global model output is used as boundary conditions in regional modelling. I have foregone anonymity as there is little to be critical of here, and I am perfectly happy for the authors to contact me informally if I can be of further assistance in making this overview as clear as possible. I fully support publication in ACP once the authors have considered these comments.

**Specific Comments**

P1 l2 Suggestion: in the title replace 'Harmonization' with 'Coordination' or 'Coordination and harmonization'.

P2 l38 …rectangular (in latitude-longitude coordinates)…

P3 l19-20 Suggest change text to: '…thus providing the opportunity of assessing the **application of these models** outside of **their** conventional modelling context.'

P3 l31 Should 'in' be 'to and from' (or 'both within and to/from')?

P3 l34 The framework **used**…

P3 l40 Capitalise Section (and subsequently p4 l38, p10 l28)

P3 l43 Insert the: '…**the** HTAP2/…'

P3 l44 Delete comma.

P4 l6-7 'were expected' -> were output

P4 l11-13 Suggest change text to: 'It **provides** specific details of the organization of the global HTAP2 and regional AQMEII3 activities, **but** only general information on the MICS3 experiments.'

P4 l16-17 Suggest change text to: 'This note **provides** coherent… characteristics to **support** the analysis…'

P4 l20 'have to be' -> were; 'a' -> this

P4 l26 Delete 'Harmonization and'. I find 'interoperability' a rather obscure word. Perhaps it is better to say (if this is what it means): 'Exchange of output data between global and regional models'?

P4 l28 'model' -> model's

P5 l8 Research

P5 l9-10 Insert bracket

P5 l15 'The so-called HTAP_v2.2 database' -> The HTAP_v2.2 emissions database

P5 l17 Clarify what emissions are used for 2009 (interpolation of 2008 and 2010?)

P5 l21 Delete 'geo-' and comma after longitude.

P6 l7 '…not necessarily consistent **with each other**,…' (as opposed to not internally consistent?)

P7 l23 deposition **fluxes**

P8 l2 …dark green colour (and with "1")…

P8 l3 delete 'colors'

P8 l13-14 ppb or ppbv (not $ppb_v$)

P9 l4 The definition of RERER (specifically the different Rs) should be clarified. My suggestion: 'where $R_{global}$ is the regional response of a quantity (e.g., surface $O_3$ concentration) in the global 20% perturbation simulation (GLO) minus the value in the unperturbed simulation (BASE); and $R_{region}$ is the regional response of that quantity in the regional 20% emission perturbation simulation minus its value in BASE.'

It may be worth giving a specific example, e.g., the RERER metric for surface $O_3$ over Europe with respect to reducing NOx emissions is derived from the EURNOX, GLONOX and BASE simulations:

$R_{global}$ = surface $O_3$ over Europe (GLONOX) minus surface $O_3$ over Europe (BASE)

$R_{region}$ = surface $O_3$ over Europe (EURNOX) minus surface $O_3$ over Europe (BASE)

P9 l15 become -> be

P10 l4-5 Delete three 'the's: '**the** examination', '**the** finer', and '**the** different'

P10 l10 Should 'in' be 'at the boundaries of'? Or maybe sometimes values throughout the box are used, rather than just values at the boundaries?

P10 l22 base -> BASE

P10 l27 GLOBALL -> GLOALL (as in Table 3?)

P10 l27-28 The inclusion of…(GLOALL) allows consistent evaluation of the RERER metric for 20% reductions of all emissions in both global and regional models.

If I am correctly understanding the RERER metric, this actually requires the NAMALL, EURALL (etc.) simulations too, so the above sentence is a bit of an oversimplification.

P11 l8 where -> were

P11 l15 Where the appropriate global model simulation boundary conditions were not available, presumably regional models just used BASE? (or GLOALL?) Or something else? This should be clarified.

P12 l5 Don't highlight '2.5'

P13 l5 Norwegian Meteorological Institute?

P15 l13-14 No italics?

P16 l15 model -> models

P16 l19 delete 'dataset of'

P16 l25 add commas after collected and geo-referenced.

P16 l36-37 Also Asian domains? (and change both -> all?)

P16 l38 Capitalise Special Issue (also p1 l40)

---

## Referee Comment (RC2) · Anonymous Referee #2 · 29 Nov 2016

Review of "Harmonization of the multi-scale multi-model activities HTAP, AQMEII and MICS-Asia: simulations, emission inventories, boundary conditions and output formats" submitted in 2016 to ACP by Galmarini et al. as a Technical Note.

General Comment

The submitted Technical Note presents the experimental scope and design of three important modelling exercises. It reviews the science and policy questions addressed

by the experiments and the key features of the technical setup. As such it will provide a very useful reference for the forthcoming studies expected to be submitted to the ACP special issue. The paper is also of good scientific quality and therefore I fully support its publication, given that the only major comment mentioned below is addressed.

Although it exceeds the mandate of the reviewer, I have an editorial question on the relevance of publishing that note in ACP whereas, for instance, all the CMIP6 model experiment description papers are published in GMD that offers to share special issues across Copernicus Journals.

Major Comment

A list of participating models for each experiment (HTAP/AQMEII/MICS-ASIA) needs to be added in order to support the level of ambition of the exercise which is stated repeatedly in the paper (ex: p3L43: "probably making HTAP2/AQMEII3/MICS3 exercise the largest, multi-scale/multi-model activity ever performed in atmospheric chemical modelling"). Given the timeline of the analysis process (with a closure of the special issue in a few months from now) all the model production is probably achieved to date. Therefore, consolidated tables of the models participating to each exercise (including the sensitivity experiments within each exercise) should be proposed, possibly adding a category as "planned contribution" if still relevant at this stage.

Minor Comments

Add version numbers in the title: HTAP2, AQMEII3, MICS-ASIA3

P2L2: Add EMEP: "The Task Force on Hemispheric Transport of Air Pollution (TF HTAP) was organized in 2005 under the Cooperative Programme for Monitoring and Evaluation of the Long-range Transmission of Air Pollutants in Europe (EMEP) of the UNECE Convention on Long-range Transboundary Air Pollution (CLRTAP) (see http://www.unece.org/env/lrtap/welcome.html)" (also in the conclusion P15L13)

P2L27: isn't the development of Integrated Assessment Tools such as HTAP-FASST

Screening Tool part of HTAP2 objectives?

P2L46: beyond the improvement in the experiment setup between HTAP 1 and 2, the concomitant ongoing improvement in model development during the past decade should also contribute to the expected advances. A paragraph on the most important development in hemispheric chemistry-transport modelling would be appreciated: e.g. what is the order of magnitude of horizontal and vertical resolution increase? How many models now include secondary inorganic and organic aerosol? Is CH4 lifetime now better constrained? (also relevant in the conclusion P16L12)

P3L3: In the CLRTAP/EMEP organisation, the regional counterpart of TF-HTAP for the European region would rather be the Task Force on Measurement and Modelling, formally speaking, it would be more appropriate to refer to AQMEII and MICS as partners of TF-HTAP.

P4L3: VOC stand for Volatile Organic Compounds

The output data formats are presented as interoperable between the global and regional portals in Section 1 (P4L8), Section 2 (P4L26), and even in the conclusion (P16L32), whereas in Section 2.5 it seems largely under development as a "pioneer" version (P13L15). Please provide more information on the design and status of this tool including the following items: Can HTAP model results be now visible in the ENSEMBLE portal, and can AQMEII data be presented in HTAP/AEROCOM tools? Does it work for all species and all models? Including grids, vertical profile, station extractions? What about interoperability with MICS outputs?

Figure 2: with emission of fire and dust left open to the modelling groups, it is unclear how the corresponding sensitivity experiment is designed.

P14L9: how are stored MICS output data: as grid or receptor based?

---

## Author Comment (AC1) · 7 Dec 2016

Response to reviewer 1 We wish to thank Dr. Stevenson for the comments and suggestions to the technical note. We agree with all suggestions and corrections proposed have been accepted and inserted and will include them in the paper.

---

## Author Comment (AC2) · 7 Dec 2016

The comment was uploaded in the form of a supplement:
http://www.atmos-chem-phys-discuss.net/acp-2016-828/acp-2016-828-AC2-supplement.pdf

---

## Author Comment (AC3) · 7 Dec 2016

We wish to thank anonymous reviewer #2 for the comments and suggestions to the technical note, which help improve its clarity. Here follows a point-by-point reply. With regard to the choice of ACP (and not GMD) for publishing this technical note and since several of the co-authors are in the Special Issue editorial team, we provide a reply, more as editors than as authors. We agree with the reviewer that the GMD would have been a good journal to publish this note. However, at the time the HTAP-AQMEII-

[Figure]

MICS-Asia special issue was initiated, we didn't foresee to have publications of this nature, and we therefore decided to not involve the GMD journal and editorial board. Unfortunately it is not possible to make this retrospectively a joint special issue either, and it would be exaggerated to do that for only one paper. Furthermore, we felt the need to link this background publication to the SI, but also recognized that this is not necessarily new science. Therefore we deliberately added 'Technical Note' in to the paper, in order not to create false expectations. Major comment. Following your request we will add tables with models, institutions for the three activities.

Minor comments:

1-Activity phase numbers have been added

2 P2L2- corrected

3 P2L27 Indeed development of HTAP-FASST or similar tools is within the objectives of the TF HTAP, as means to integrate science in user-friendly tools for scenario assessments and policy advise.

4 P2L46 These issues are going to be addressed by the individual model developers and user when describing the model versioning in the individual publications that will constitute the special issue or within a preface or concluding publication for the special issue. The point is very important and deserves a a more extended discussion in individual and overview papers rather than in this technical note. However, we do propose to include the follow sentence: * Since the HTAP1 experiments models have been updated with newer parameterisations, include higher resolutions, and include more components. However, analysis within HTAP2 and other modeling activities, will have to demonstrate whether this has resulted in better constrained model results, and sensitivities to key-processes.

5 P3L3 We agree that in the policy framework TF HTAP is more looking at hemispheric/global issues, whereas the TF MM (Models and Measurements) is looking at

regional (European) model/measurement issues- and in fact the two Task Forces collaborate on joint issues. We intended really to talk about the modeling experiments. A correction has been made in the text now referring to HTAP2 and the regional counterpart AQMEII3, rather than TF HTAP.

6 P4L3- corrected

7 As the reviewer probably realizes, interoperability is a goal worth pursuing, but with several hurdles on the road. HTAP and AQMEII communities have made important steps forward. The HTAP and ENSEMBLE formats can be converted into one another by means of dedicated software, thus allowing for consultability of the results by the two communities. HTAP2 results have been imported into the ENSEMBLES platform, but the AQMEII results have not been included in the MetNo platform yet. Lack of resources and time has prevented the full development of interoperability of AEROCOM and ENSEMBLE though in essence that can be realized technically once the data can be converted reciprocally into each other format. Interoperability in MICS have been somewhat slower, but we hope that this exercise will help to facilitate progress in interoperability for MICS as well. The ultimate goal would be internationally agreed formats, with all meta-data included, and easy to implement (the latter is still a problem). The text has been streamlined with respect to this clarification point.

8- Figure 2. As mineral dust (and seasalt) are nowadays mostly interactively calculated in global models, it was considered to be a step-back by most modellers to use prescribed emissions (as was for instance done in AEROCOM phase 1). Regarding biomass burning emissions, we recommended GFED3, but explicitly allowed modelers to use also alternative inventories, recognizing the fact that currently there is no single biomass burning inventory that has been accepted as having outstanding performance with respect to others. Most of the model experiments do not focus on biomass/dust; however in the joint AEROCOM3 work, there are specific analyses of aerosol with regard to model' sensitivities to emissions.

9 P14L19- Both